# Controlling shareholders' share pledge and share repurchase notices of listed companies

Limei Cao [1]*, Xiaoyun Zhu[2], Jiani Xie[3]

1 School of Accounting, Guangdong University of Finance and Economics, Guangzhou, Guangdong, China,
2 School of Accounting, Guangzhou Huashang College, Guangzhou, Guangdong, China, 3 Ganzhou Urban Investment Holding Group CO., LTD, Ganzhou, Jiangxi, China

* caolimei925@163.com

## Abstract

States encourage listed companies to use stock repurchase to elevate the market value of listed firms. After China's promulgation of the new Company Law in 2018, the number of listed companies that issued stock repurchase notices has increased, and the frequency is also increasing. But whether market value management is the real incentive for the action remains debatable. To reduce the risk of pledges, controlling shareholders may use stock repurchases to maintain the security of control rights, and stock repurchase notice may become a tool for controlling shareholders to manage pledge risks. From the perspective of pledge risk management, this paper selects the listed companies from 2012 to 2019 and finds that the share pledge of the controlling shareholder affects the stock repurchase behavior of listed companies by affecting the current pledge risk and the quality of information disclosure plays the interactive role between the two.

## 1. Introduction

The use of share buybacks by companies can send a signal to the outside world that share prices are undervalued, improve the capital structure of the company, and increase enterprise value [1]. The more confident the manager is to the enterprise, the higher the probability of stock repurchase of the listed company. But the prevalence of agency problems makes it possible that share repurchases may be alienated as a way for insiders to manipulate earnings [2], controlling shareholders and executives may also issue stock repurchase alerts for personal gain and encroachment on the company, and companies are more likely to buy back shares when executive compensation is linked to earnings per share [3]. Li and He took the case of share repurchase in the open market of our country's stock market after 2005 as the research object, and find that the information content in the repurchase announcement conforms to the undervaluation hypothesis, there are opportunities to achieve large shareholders arbitrage phenomenon [4]. Share repurchase announcements are accompanied by a reduction in long-term stock returns and insider selling [5].

In recent years, the stock market in China has been turbulent. As one method of capital operation, stock repurchases can effectively stabilize the stock prices. To encourage listed companies to use stock repurchase for market value management, China's new Company Law in

**Data Availability Statement:** All relevant data are within the article.

**Funding:** This work was supported by the Guangdong Provincial Department of Education Innovation Team Project "Big Data Audit Research

Team" (2022WCXTD009), the General Project of Philosophy and Social Science Planning in Guangdong Province (NO.GD23CGL27), and the General Projects of the National Social Science Fund (23BGL053).

**Competing interests:** The authors have declared that no competing interests exist.

2018 included market value management in the scope of allowing listed companies to repurchase shares. As a result, there was a rapid rise in listed companies that issued stock repurchase forecasts in 2018. As an important sign to the outside world as to the intention of listed companies, the share repurchase notice signals that the company's stock price is undervalued, which can support and stabilize the company's stock price [1]. But the positive impact of share repurchase forecast on stock prices may also make share repurchase a tool to manipulate stock prices. Whether the pledge of share will prompt the listed company to issue the share repurchase notice and the real purpose of the share repurchase notice under the background of the share pledge remains to be explored.

This paper uses China's A-share listing from 2012 to 2019 as the sample for research. The main issue of this paper is: Will the pledge of majority shareholders increase the probability that the listed companies will issue share repurchase plans to convey the signal of underpricing to the market, and how will the market react to the repurchase plans? After the release of the plan, whether the pledge of major shareholders will affect the company's ultimate repurchase implementation? Will the execution of share repurchases affect the long-term return of the company's stock? This paper intends to explore these issues in depth. It was found that listed companies with controlling shareholder pledge their share have a greater probability of issuing share repurchase notice. The higher the pledge ratio, the greater the likelihood that listed companies will issue share repurchase notice. The quality of information disclosure will affect the relationship between equity pledges and share repurchase of the controlling shareholders. In further analysis, this paper verifies the positive impact of the share repurchase forecast on the stock price under the background of equity pledge. Under the background of the equity pledge of the controlling shareholder, the corporate share repurchase will harm the business performance and the company value. The share repurchase notice may be a means for the controlling shareholder to deal with the pledge risk. This paper explores the relationship between controlling shareholders' equity pledges and stock repurchase notice from the perspective of pledge risk management, analyzes the action mechanism between the two, and expands the relevant research on the economic consequences of controlling shareholders' equity pledges.

This research found that the share pledge of the controlling shareholder is the motivation for the listed company to issue the share repurchase notice. When the controlling shareholder has the share pledge behavior or the proportion of the share pledge is higher, the probability of the listed company issuing the stock repurchase announcement increases significantly. The relationship will be affected by the company's information transparency; with lower company information transparency, the relationship between the company's equity pledge and the company declared stock repurchase is more significant. From the perspective of maintaining the control and safety of major shareholders, this paper expands the relevant research on the motivation of the stock repurchase announcement of listed companies. In addition, it enriches the discussion of the economic consequences of major shareholders' equity pledges.

## 2. Theoretical analysis and research hypothesis

In recent years, the share pledge has increasingly become an important financing method for the controlling shareholders of listed companies. Although the share pledge procedures are simple than the real estate and chattel mortgages, the pledge also affects the property rights of the controlling shareholders.

In the case of the high stock concentration of listed companies in China, the share pledge of the controlling shareholders is more common, and the controlling shareholder share pledge may motivate the listed companies to issue the share repurchase notice. When the controlling shareholder pledges shares, the contract often establishes a cordon, flat line, and other terms to

protect the interests of creditors. During the period of share pledge of the controlling shareholder, once the stock price falls on the warning line or flat line, if the controlling shareholder cannot make a margin call in time, the pledgee will sell the shares pledged by the shareholder, and the listed company will risk a transfer of control. This scenario forces the controlling shareholders to focus on, stabilize, or raise their share prices. For instance, the controlling shareholders at the meeting signed performance commitments [6] and implemented charitable donations and "high transfer" policies [7, 8]. In addition, they may release "good news" to external investors and engage in selective disclosure of company information or otherwise hidden "bad news" [9]. Companies that pledge their controlling shareholders also develop real surpluses by manipulating surplus and capitalizing through R&D spending [10]. When the margin pressure is relatively large, the holding shareholders will maintain their control through share repurchase and other means [11].

Compared with the above methods, a share repurchase forecast has obvious advantages in helping control shareholders and alleviate share pledge risks. First, share repurchase notice can stabilize and improve the company's stock price in the short term. It has a positive impact on the company's share price, and the stock excess return rate is significantly positive on the day of the stock notice announcement [12]. The stock repurchase notice can increase the stock price and reduce the pledge risk of bursting and liquidation. Second, the company can choose the repurchase implementation point after the share repurchase notice, and the controlling shareholder can complete the repurchase plan within the specified time limit. After announcing the repurchase notice, the controlling shareholders shall have enough time to find other ways to raise the margin or supplements. Third, the control of the controlling shareholder does not change, and they still enjoy the ability to influence the company's decisions through the shareholders' meetings. The controlling shareholder can use its control over the company to encourage the company to issue a share repurchase notice. Fourth, the release of share repurchase notice cost is low, and the procedures are relatively simple. Under the current laws and regulations, the listed company declares that the share repurchase plan only requires the board of directors' resolution. This paper makes the following assumptions:

**Hypothesis 1**: **Under certain other conditions, the share pledge of the controlling shareholder will prompt the listed company to issue a share repurchase notice. The higher the proportion of share pledges, the more likely the listed company will issue a share repurchase notice**.

Information asymmetry is the fundamental reason that the company's share price cannot correctly reflect the company's internal value [13]. Therefore, the quality of information disclosure of listed companies may affect the relationship between the controlling shareholder's share pledge and the share repurchase notice.

Since the capital market has not reached the ideal state of strong and effective market efficiency, there is a deviation between the market value and the internal value of listed companies. After the share pledge, investor panic and concerns widened the negative gap between the stock prices of listed companies and their real value. The worse the quality of information disclosure, the more obvious the negative impact of the controlling shareholder's equity pledge on the investment sentiment. In the background of an equity pledge, information asymmetry magnifies the abnormal fluctuation degree of the company's stock price and expands the risk of the stock price collapse in the future [14]. To alleviate the risk of control transfer caused by stock price deviation or collapse, listed companies and their insiders have a strong incentive to issue share repurchase notice during the equity pledge.

The lower the quality of company information disclosure, the more difficult it is for external investors to have confidence that the company's information is genuine. When a listed

company issues a share repurchase notice, investors cannot distinguish the real cause of the company's share repurchase notice because of information asymmetry, so it is easy to interpret the company's repurchase notice as the listed company is undervalued to the outside world and to therefore buy the company's shares blindly. China's capital market speculative atmosphere is relatively strong, many retail investors lack rational analysis abilities, and the phenomenon of "following the crowd" is obvious. Under this "herd effect," the impact of the stock repurchase forecast on investors will be amplified. The worse the quality of information disclosure, the more obvious the market effect of the repurchase notice [15]. The higher the degree of information asymmetry, the greater the pledge risk faced by listed companies and their insiders, the stronger the positive market effect generated by stock repurchase, and the higher the probability of controlling shareholders using stock repurchase to reduce the pledge risk. As a consequence, we hypothesize that:

**Hypothesis 2**: **Under certain other conditions, the worse the quality of information disclosure, the stronger the positive relationship between the controlling shareholder's share pledge and issuing stock repurchase notice by the listed company**.

## 3. Research design

### 3.1 Samples and data sources

Before 2012, Chinese companies needed to approve their shares, and the number of listed companies that issued share repurchase announcements was small. China's economy, stock market and corporate repurchase activity were frequently shut down in 2020 due to the covid-19 pandemic, so the sample period did not include 2020 and beyond, This paper selects Chinese A-share listed companies as the research object and selects the research sample from 2012 to 2019 to explore the relationship between the controlling shareholder's equity pledge and share repurchase notice. This article refers to the WIND database of the stock repurchase forecast, which consists of six categories: market value management, equity incentive, equity incentive cancellation (including equity incentive plan during the enterprise performance and the original equity incentive plan employees leave equity incentive cancellation), profit compensation, equity incentive plan termination, and reorganization. The cancellation of equity incentives, profit compensation, termination of the equity incentive plan, and reorganization are all affected by external factors with relatively small influence by the controlling shareholders. Therefore, the purpose of the repurchase is to exclude the data related to equity incentive cancellation, profit compensation, termination of the equity incentive plan, and reorganization. This paper also excluded the sample firms in the financial industry, ST or * ST company samples, and observations with missing variables. Among them, we completed the relevant data of the stock repurchase forecast through manual collection and supplement, and the other data were drawn from the national CSMAR database. To reduce the effect of extreme values, we trimmed the main continuous variables on the 1% and 99% percentiles, resulting in 12,980 observations used in the following analysis.

### 3.2 Variable definitions and descriptions

**3.2.1 Share repurchase notice.** Whether the listed company can issue the stock repurchase notice is affected by the controlling shareholders, so we can divide the stock repurchase notice as either an active or passive repurchase.

An active repurchase refers to the stock repurchase behavior carried out by the company independently because of its development, mainly including market value management and

equity incentives. Passive repurchase means the stock repurchase that must be carried out, mainly including equity incentives and cancellation during the equity incentive plan period (including equity incentive cancellation because of the failure of the enterprise performance and the resignation of the original equity incentive plan), profit compensation, termination of equity incentive plan and asset restructuring. Passive repurchases are more greatly affected by the external environment, with a low controllable degree, and relatively less affected by the subjective will of the controlling shareholders. Therefore, this paper mainly discusses whether the probability of using the stock repurchase forecast to increase the stock price rises after the share pledge and whether the passive repurchase can reflect the subjective will of the controlling shareholder. As a result, only the two types of active repurchase with the purpose of market value management and equity incentive are retained as the research object [16–20]. Drawing on Huang and Wang [21], this paper measures the release of the current stock repurchase notice of listed companies by constructing the dummy variable Rp.

**3.2.2 Share pledge of the controlling shareholder (PLD, PLR).** Referring to the definitions provided by Deren et al. [22], this paper constructs the virtual variable PLD and the continuous variable PLR to measure the share pledge of the controlling shareholders. If the controlling shareholder of the enterprise has share pledge, the PLD is 1; otherwise, it is 0. PLR is equal to the proportion of the number of shares pledged by the controlling shareholder in the number of listed companies held by it.

**3.2.3 Quality of information disclosure.** The KV index was calculated by model (1) in reference to Xu and Xu [13]. The $P_t$ and $Vol_t$ are found in the model (1) and are the closing price and trading volume (number of shares) of day t. $Vol_0$ is the average daily trading volume of all trading days during the study period. Formulae (1) is calculated using least squares regression (excluding values below zero). The better the quality of information disclosure of listed companies, the easier it is for investors to evaluate the value of listed companies based on information disclosure (including both compulsory and voluntary information disclosure). The lower the dependence on stock trading volume, the weaker the relationship between stock trading volume and investor return rate. On the contrary, the worse the quality of information disclosure, the more investors need to judge the investment value of listed companies with the help of stock trading volume, the stronger investors rely on stock trading volume, and the relationship between stock trading volume and investor yield is stronger. The larger the KV index, the worse the information disclosure.

$$Ln|(P_t - P_{t-1})/P_{t-1}| = \lambda_0 + \lambda(Vol_t/Vol_0 - 1) + \varepsilon \tag{1}$$

## 3.3 Control variables

Dittmar [23] and Lin [24] also control company size (Size), asset-liability ratio (Lev), return on assets (ROA), book value ratio (BM), largest shareholder shareholding (Top1), institutional investor shareholding (INST), integration (Dual), corporate free cash flow (FCF), year of listing (LAG), board size (Board), industry fixed effect (Ind), and annual fixed effect (Year). The specific definition and calculation process of the variables are shown in Table 1.

## 3.4 Model design

To test Hypothesis 1, we constructed a regression model (2), where Rp is the dependent variable, PLD and PLR are the independent variables, and *Controls* represent all control variables.

**Table 1. Definition of variables.**

|  | Variable | Definition |
|---|---|---|
| Dependent variable | Rp | Dummy variable which will equal 1 if the company issued a share repurchase notice, otherwise it will be 0 |
| Independent variable | PLD | Dummy variable, which will equal 1 if the controlling shareholder of the company pledge their share, otherwise it will be 0. |
|  | PLR | The ratio of the amount of share pledged by the controlling shareholder over the total amount of share held by the controlling shareholder of the listed company |
| Moderator variable | KV | According to the method of Xu Shoufu and Xu Longbing (2015), and the specific calculation method, see the above |
| Control variable | Size | Natural logarithm of the total company assets at the end of the term |
|  | Lev | The ratio of total liabilities over total assets |
|  | ROA | The ratio of net profit over total assets |
|  | BM | The ratio of total assets over the company market value |
|  | Top1 | Shareholding ratio of the company's largest shareholder |
|  | INST | The ratio of the amount of share owned by the corporate investors over the total amount of share held by the controlling shareholder of the listed company |
|  | DIV | The ratio of dividend per share over earnings per share |
|  | Dual | Dummy variable will be 1 if the chairman of the company concurrently serves as general manager, otherwise it will be 0 |
|  | FCF | The ratio of the free cash flow of the company over total assets |
|  | LAG | The natural log of calculated result of the current year minus the year of listing plus one |
|  | Board | The natural log of the number of directors |
|  | Ind | Industry dummy |
|  | Year | Year dummy |

In model (2), the coefficient of the significant timing assumption holds.

$$Rp_{i,t} = \beta_0 + \beta_1 PLD_{i,t-1}/PLR_{i,t-1} + Controls_{i,t-1} + \sum Year + \sum Ind + \varepsilon_{i,t-1} \qquad (2)$$

To test the adjustment effect of information disclosure quality, we build models (3) and models (4). Among them, KV is the measure of information disclosure quality, and BLC is the measure of equity checks and balances. The larger the KV, the worse the quality of information disclosure. The regression results in 3 in the model (3) and model (4).

$$Rp_{i,t} = \beta_0 + \beta_1 PLD_{i,t-1}/PLR_{i,t-1} + \beta_2 KV_{i,t-1} + \beta_3 KV_{i,t-1} \times PLD_{i,t-1}/PLR_{i,t-1}$$
$$+ Controls_{i,t-1} + \sum Year + \sum Ind + \varepsilon_{i,t-1} \qquad (3)$$

## 4. Empirical results

### 4.1 Descriptive statistics

Table 2 presents the descriptive statistics table of the main variables. It can be seen from Table 2 that the average value of share repurchase notice Rp is 0.051, and the variance is 0.220. The sample size of companies issuing the share repurchase notice only accounts for 5.1% of the total sample. There are significant differences in the release of share repurchase notices between companies. The average PLR ratio of PLD share pledge and controlling shareholder share pledge is 0.410 and 0.223, respectively, and the standard deviation is 0.492 and 0.327, respectively, indicating the large difference in share pledge of controlling shareholders of

**Table 2. Descriptive statistics.**

| Variable Name | N | Mean | Variance | Max | Min |
|---|---|---|---|---|---|
| Rp | 12980 | 0.051 | 0.220 | 1 | 0 |
| PLD | 12980 | 0.410 | 0.492 | 1 | 0 |
| PLR | 12980 | 0.223 | 0.327 | 1.000 | 0.000 |
| KV | 12980 | 0.495 | 0.195 | 1.978 | 0.001 |
| Size | 12980 | 22.250 | 1.282 | 26.250 | 19.52 |
| Lev | 12980 | 0.407 | 0.200 | 0.925 | 0.035 |
| ROA | 12980 | 0.055 | 0.042 | 0.222 | 0 |
| BM | 12980 | 0.984 | 1.075 | 8.061 | 0.051 |
| Top1 | 12980 | 0.357 | 0.148 | 0.758 | 0.084 |
| INST | 12980 | 0.409 | 0.235 | 0.889 | 0.000 |
| DIV | 12980 | 0.328 | 1.261 | 107.400 | 0.000 |
| Dual | 12980 | 0.265 | 0.441 | 1 | 0 |
| FCF | 12980 | 0.013 | 0.102 | 1.083 | -0.935 |
| LAG | 12980 | 2.113 | 0.782 | 3.296 | 0.693 |
| Board | 12980 | 2.137 | 0.194 | 2.708 | 1.609 |

different enterprises. The average KV of information disclosure quality is 0.495, the minimum value is 0.001, and the maximum value is 1.978, indicating that the information disclosure quality gap between enterprises is large. There is also a large gap in the control variables such as size, book market value ratio of BM, and enterprise free cash flow (FCF). The average Top1 shareholding ratio of the largest shareholder of the enterprise is 0.357, and the maximum value is as high as 0.758, indicating the high degree of equity concentration of the company.

In this paper, the data are classified according to whether the controlling shareholders have share pledges, and the mean indexes of the two groups are evaluated. The results are shown in Table 3. As can be seen from Table 3, the probability of the stock repurchase notice Rp of listed companies where the controlling shareholder does not pledge equity is 0.027, and the stock repurchase notice of the listed company where the share pledge is 0.084. The probability of a stock repurchase notice issued by the latter is more than twice that of the former. The results of the t-test for the group means showed that the difference in the above proportions was significant at the 1% level. The average information disclosure quality KV of listed companies without equity pledge behavior are 0.491, and the average sample information disclosure quality KV of controlling shareholders without equity pledge behavior of controlling shareholders is 0.500. The ROA of the enterprises pledged by the controlling shareholder and those not pledged by the controlling shareholder is 0.055, and the difference between DIV between the two groups of samples is not significant.

## 4.2 Analysis of the empirical results

The model (2) regression results are shown in Table 4. Since STATA deleted the observations with a single independent variable during the binary selection model estimation, the regression sample for this section contains 12,944 observations. Columns (1) and (2) in Table 4 are listed as the regression results without adding control variables. Column (3) and column (4) control the size of the enterprise, asset-liability ratio, return on assets, and other indicators. The share repurchase forecast Rp, share pledge PLD and share pledge ratio PLR are significantly positive at the 1% level, indicating that companies with controlling shareholder share pledge are more likely to issue share repurchase notice, and the higher the share pledge ratio, the more likely

**Table 3. T test of difference of descriptive statistics of sample firms.**

| Variable Name | Controlling shareholder does not pledge share | | Controlling shareholders pledge share | | t-test |
|---|---|---|---|---|---|
| | N | Variable | N | Mean | |
| Rp | 7654 | 0.027 | 5326 | 0.084 | -0.057*** |
| PLR | 7654 | 0.000 | 5326 | 0.544 | -0.544*** |
| KV | 7654 | 0.491 | 5326 | 0.500 | -0.010*** |
| BLC | 7654 | 0.678 | 5326 | 0.743 | -0.065*** |
| Size | 7654 | 22.360 | 5326 | 22.070 | 0.291*** |
| Lev | 7654 | 0.408 | 5326 | 0.405 | 0.003 |
| ROA | 7654 | 0.055 | 5326 | 0.055 | 0.000 |
| BM | 7654 | 1.081 | 5326 | 0.844 | 0.237*** |
| Top1 | 7654 | 0.369 | 5326 | 0.339 | 0.030*** |
| INST | 7654 | 0.440 | 5326 | 0.364 | 0.076*** |
| DIV | 7654 | 0.317 | 5326 | 0.345 | -0.028 |
| Dual | 7654 | 0.211 | 5326 | 0.342 | 0.131*** |
| FCF | 7654 | 0.016 | 5326 | 0.008 | 0.008*** |
| LAG | 7654 | 2.206 | 5326 | 1.978 | 0.228*** |
| Board | 7654 | 2.163 | 5326 | 2.099 | 0.064*** |

Note: The sample is divided into two groups according to whether controlling shareholder of the listed company pledge their share, and T test is conducted to test whether mean of variables between these two groups if significant.

*** is significant at 1%;

** is significant at 5%;

* is significant at 10%.

listed companies are to issue share repurchase notice. Empirical results support hypothesis 1. In terms of control variables, the larger the enterprise's assets, the higher the dividend distribution ratio, and the greater the probability of stock repurchase. Debt financing will reduce the probability of companies announcing buyback plans. The regression results are basically the same prediction as before.

In this paper, the KV index is calculated according to the method of Xu and Xu (2015). The larger the KV index, the worse the information disclosure quality. The regression results of Hypothesis 2 are shown in Table 5.

It can be seen from Table 5 that PLD, PLR and KV deliveries are significantly positive at 1% and 5%, respectively, which verifies Hypothesis 2, that is, the lower the quality of information disclosure is, the greater the probability of issuing stock repurchase notice by a company with share pledge of the controlling shareholder.

### 4.3 Robustness tests

**4.3.1 Tool variable method.** To reduce the impact of endogenous problems on the regression results, this paper selects the industry average pledge rate (Pledge_ip) in the province where the company is located as the tool variable. Referring to the studies of Chen et al. [25] and Ma et al. [26] in this paper, the OLS model is used to test the effectiveness of the tool variables. The specific regression results are shown in Table 6. The regression results showed that Pledge_ip was associated with PLD and PLR, but not with Rp, and the instrumental variables basically met the selection requirements. This article uses a two-stage regression model to test the empirical results, and the regression results are shown in Table 7. According to Table 7, both the AR test and Wald test are significant at the 1% level, indicating that there is no weak

**Table 4. Controlling shareholders' stock pledge and share repurchase of listed firms.**

| | (1) | (2) | (3) | (4) |
|---|---|---|---|---|
| | Rp | Rp | Rp | Rp |
| PLD | *0.890*** | | *0.962*** | |
| | *(9.859)* | | *(9.783)* | |
| PLR | | *1.088*** | | *1.213*** |
| | | *(9.887)* | | *(10.283)* |
| Size | | | 0.585*** | 0.592*** |
| | | | (10.543) | (10.676) |
| Lev | | | -1.845*** | -1.792*** |
| | | | (-5.650) | (-5.548) |
| ROA | | | 4.817*** | 5.319*** |
| | | | (4.826) | (5.400) |
| BM | | | -0.109 | -0.119* |
| | | | (-1.562) | (-1.678) |
| Top1 | | | -2.197*** | -2.086*** |
| | | | (-6.185) | (-5.966) |
| INST | | | -0.203 | -0.295 |
| | | | (-0.838) | (-1.228) |
| DIV | | | 0.046*** | 0.046*** |
| | | | (3.804) | (3.497) |
| Dual | | | 0.185* | 0.192** |
| | | | (1.903) | (1.972) |
| FCF | | | -0.995** | -1.117*** |
| | | | (-2.302) | (-2.609) |
| LAG | | | -0.127* | -0.200*** |
| | | | (-1.872) | (-3.005) |
| Board | | | -0.338 | -0.345 |
| | | | (-1.454) | (-1.481) |
| Ind | Yes | Yes | Yes | Yes |
| Year | Yes | Yes | Yes | Yes |
| _cons | -5.583*** | -5.472*** | -16.360*** | -16.313*** |
| | (-9.367) | (-9.304) | (-12.104) | (-12.144) |
| N | 12944 | 12944 | 12944 | 12944 |
| Wald_Chi2 | 690.53*** | 701.44*** | 923.82*** | 937.28*** |
| Pseudo_R2 | 0.205 | 0.202 | 0.248 | 0.246 |

Note: The t-statistic is shown in parentheses.

*** is significant at 1%;

** is significant at 5%;

* is significant at 10%.

instrumental variable issue. In the second stage of regression, PLD and PLR are significantly positive at the 1% level, indicating that the pledge of controlling shareholder equity does indeed increase the probability of listed companies issuing stock repurchase notices.

**4.3.2 Checking the regression model results.** To evaluate the empirical results' robustness, we used the probit model, shown in Table 8. The regression results were essentially consistent with our earlier results, and the study hypothesis was verified.

**Table 5. The interactive effect of the quality of information disclosure and ownership structure.**

| | (1) | (2) |
|---|---|---|
| | Rp | Rp |
| PLD | 0.327 | |
| | (1.368) | |
| PLR | | 0.451 |
| | | (1.428) |
| KV | -1.046*** | -0.772** |
| | (-2.826) | (-2.483) |
| PLD×KV | *1.197*** | |
| | *(2.788)* | |
| PLR×KV | | *1.440*** |
| | | *(2.540)* |
| BLC | | |
| PLD×BLC | | |
| PLR×BLC | | |
| Size | 0.614*** | 0.621*** |
| | (10.496) | (10.644) |
| Lev | -1.862*** | -1.789*** |
| | (-5.678) | (-5.531) |
| ROA | 5.118*** | 5.676*** |
| | (4.994) | (5.611) |
| BM | -0.130* | -0.141* |
| | (-1.810) | (-1.924) |
| Top1 | -2.199*** | -2.098*** |
| | (-6.134) | (-5.962) |
| INST | -0.174 | -0.271 |
| | (-0.713) | (-1.115) |
| DIV | 0.047*** | 0.048*** |
| | (3.874) | (3.755) |
| Dual | 0.191** | 0.198** |
| | (1.961) | (2.026) |
| FCF | -0.980** | -1.118*** |
| | (-2.262) | (-2.596) |
| LAG | -0.125* | -0.200*** |
| | (-1.843) | (-3.000) |
| Board | -0.350 | -0.350 |
| | (-1.506) | (-1.505) |
| Ind | Yes | Yes |
| Year | Yes | Yes |
| cons | -16.519*** | -16.591*** |
| | (-12.062) | (-12.186) |
| N | 12944 | 12944 |
| WaldChi2 | 935.41*** | 942.82*** |
| PseudoR$^2$ | 0.249 | 0.247 |

Note: The t-statistic is shown in parentheses.

*** is significant at 1%;

** is significant at 5%;

* is significant at 10%.

**Table 6. Test of instrumental variable selection.**

| | (1) | (2) | (3) | (4) |
|---|---|---|---|---|
| | Rp | Rp | PLD | PLR |
| PLD | 0.042*** | | | |
| | (8.645) | | | |
| PLR | | 0.065*** | | |
| | | (7.745) | | |
| Pledge_ip | 0.006 | -0.007 | 1.183*** | 0.974*** |
| | (0.483) | (-0.585) | (82.838) | (110.921) |
| Size | 0.026*** | 0.026*** | -0.009** | -0.005** |
| | (10.029) | (9.996) | (-1.976) | (-1.987) |
| Lev | -0.063*** | -0.062*** | 0.277*** | 0.157*** |
| | (-4.958) | (-4.827) | (10.905) | (9.531) |
| ROA | 0.269*** | 0.288*** | -0.222** | -0.429*** |
| | (4.973) | (5.309) | (-2.265) | (-7.068) |
| BM | -0.008*** | -0.008*** | -0.030*** | -0.019*** |
| | (-2.824) | (-2.827) | (-5.728) | (-5.893) |
| Top1 | -0.086*** | -0.082*** | -0.075*** | -0.108*** |
| | (-6.515) | (-6.223) | (-2.856) | (-6.672) |
| INST | -0.012 | -0.014 | -0.054*** | 0.006 |
| | (-1.191) | (-1.468) | (-2.808) | (0.501) |
| DIV | 0.003 | 0.003 | 0.003 | 0.002 |
| | (1.128) | (1.089) | (1.333) | (1.184) |
| Dual | 0.011** | 0.011** | 0.069*** | 0.037*** |
| | (2.252) | (2.368) | (7.847) | (6.581) |
| FCF | -0.041** | -0.044** | -0.148*** | -0.061** |
| | (-2.228) | (-2.347) | (-4.118) | (-2.434) |
| LAG | -0.001 | -0.003 | -0.070*** | -0.006* |
| | (-0.201) | (-1.116) | (-12.831) | (-1.880) |
| Board | -0.017 | -0.016 | -0.144*** | -0.100*** |
| | (-1.607) | (-1.567) | (-7.532) | (-8.342) |
| Ind | Yes | Yes | Yes | Yes |
| Year | Yes | Yes | Yes | Yes |
| _cons | -0.503*** | -0.498*** | 0.662*** | 0.348*** |
| | (-9.222) | (-9.121) | (7.148) | (6.073) |
| N | 12980 | 12980 | 12980 | 12980 |
| $R^2$ | 0.107 | 0.106 | 0.355 | 0.429 |

Note: The t-statistic is shown in parentheses.

*** is significant at 1%;

** is significant at 5%;

* is significant at 10%.

**4.3.3 Heckman two-stage model.** To reduce the adverse effects of sample self-selection and endogeneity, the Heckman two-stage model was used to test Hypotheses 1, 2, and 3. This paper first estimates the influence of company size (Size), asset-liability ratio (Lev), return on assets (ROA) and other control variables on share pledge (PLD) and then takes the calculated inverse Mills ratio (IMR) as a control variable into formulas (2) to (4). Table 9 shows the regression results of the two-stage Heckman regression. It can be found from columns (2) and

**Table 7. Two-stage regression.**

| | (1) | (2) | (3) | (4) |
|---|---|---|---|---|
| | First | Second | First | Second |
| | RLD | Rp | PLR | Rp |
| Pledge_ip | *1.183*\*\*\* | | *0.974*\*\*\* | |
| | *(66.194)* | | *(87.146)* | |
| PLD | | *0.556*\*\*\* | | |
| | | *(6.023)* | | |
| PLR | | | | *0.617*\*\*\* |
| | | | | *(5.591)* |
| Controls | Yes | Yes | Yes | Yes |
| _cons | 0.662\*\*\* | -8.468\*\*\* | 0.348\*\*\* | -1.397\*\*\* |
| | (6.614) | (-12.577) | (5.557) | (-12.516) |
| Ind | Yes | Yes | Yes | Yes |
| Year | Yes | Yes | Yes | Yes |
| N | 12980 | 12940 | 12980 | 12940 |
| AR | 115.05\*\*\* | | 103.18\*\*\* | |
| Wald | 115.83\*\*\* | | 103.54\*\*\* | |

Note: The t-statistic is shown in parentheses.

\*\*\* is significant at 1%;

\*\* is significant at 5%;

\* is significant at 10%.

(3) that the relationship between the controlling shareholder's share pledge and the share repurchase notice is still significantly positive. From columns (4) and (5), the coefficients of PLD KV and PLR KV are significantly positive. The results indicate that the present study conclusions are robust and not influenced by endogeneity.

**4.3.4 Propensity matching score method.** This paper uses the propensity score matching method (PSM) to solve the endogenous problem and the nearest neighbor matching (1:1) method. Specifically, this paper selects whether the company has a controlling shareholder share pledge as the indicator variable through the company size (Size), asset-liability ratio (Lev), return on assets (ROA), book value ratio (BM), the largest shareholder shareholding (Top1), institutional investor shareholding (INST), two job integration degree (Dual), company listing year (LAG), board size (Board) and other indicators, and constructs a logistic model to calculate the tendency score. The results of the propensity score matches are shown in Table 10.

Second, the balance trend test was conducted, and the specific results are shown in Table 11. The variables in the table are basically below 5% after matching.

Table 12 reports the regression results after matching samples, and the empirical results are essentially consistent with the previous study assumptions.

**4.3.5 Excluding observations from 2015.** In 2015, China suffered a large-scale stock market crash, and many stocks were suspended, which might impact the company's stock repurchase decision. Therefore, this paper excluded observations from 2015 and then re-tested the three hypotheses, and the test results are shown in Table 13. It can be seen from Table 13 that companies with share pledge of the controlling shareholder have a greater probability of issuing a share repurchase announcement; the worse the quality of information disclosure, the greater the probability of issuing a share repurchase announcement. The regression results are

**Table 8. Robustness check of different regression models.**

|  | (1) | (2) | (3) | (4) |
|---|---|---|---|---|
|  | Rp | Rp | Rp | Rp |
| PLD | 0.470*** |  | 0.134 |  |
|  | (10.007) |  | (1.122) |  |
| PLR |  | 0.615*** |  | 0.188 |
|  |  | (10.207) |  | (1.155) |
| KV |  |  | -0.448** | -0.317** |
|  |  |  | (-2.455) | (-2.020) |
| PLD ×KV |  |  | 0.642*** |  |
|  |  |  | (2.947) |  |
| PLR×KV |  |  |  | 0.817*** |
|  |  |  |  | (2.772) |
| BLC |  |  |  |  |
| PL ×BLC |  |  |  |  |
| PL×BLC |  |  |  |  |
| Size | 0.303*** | 0.305*** | 0.310*** | 0.313*** |
|  | (10.669) | (10.783) | (10.441) | (10.555) |
| Lev | -0.925*** | -0.899*** | -0.929*** | -0.899*** |
|  | (-5.667) | (-5.556) | (-5.681) | (-5.552) |
| ROA | 2.711*** | 2.924*** | 2.802*** | 3.026*** |
|  | (5.186) | (5.647) | (5.233) | (5.702) |
| BM | -0.046 | -0.050 | -0.051 | -0.056 |
|  | (-1.351) | (-1.464) | (-1.480) | (-1.589) |
| Top1 | -1.114*** | -1.052*** | -1.105*** | -1.044*** |
|  | (-6.160) | (-5.883) | (-6.074) | (-5.806) |
| INST | -0.106 | -0.145 | -0.099 | -0.141 |
|  | (-0.877) | (-1.200) | (-0.807) | (-1.164) |
| DIV | 0.023*** | 0.023** | 0.024*** | 0.024*** |
|  | (2.729) | (2.540) | (2.817) | (2.753) |
| Dual | 0.100** | 0.106** | 0.102** | 0.108** |
|  | (2.033) | (2.151) | (2.070) | (2.187) |
| FCF | -0.463** | -0.523** | -0.453** | -0.521** |
|  | (-2.111) | (-2.400) | (-2.066) | (-2.388) |
| LAG | -0.070** | -0.105*** | -0.069** | -0.103*** |
|  | (-2.062) | (-3.128) | (-2.019) | (-3.098) |
| Board | -0.182 | -0.180 | -0.182 | -0.179 |
|  | (-1.553) | (-1.536) | (-1.556) | (-1.530) |
| Ind | Yes | Yes | Yes | Yes |
| Year | Yes | Yes | Yes | Yes |
| cons | -8.364*** | -8.354*** | -8.346*** | -8.388*** |
|  | (-12.53) | (-12.58) | (-12.37) | (-12.48) |
| N | 12944 | 12944 | 12944 | 12944 |
| WaldChi2 | 930.53*** | 937.90*** | 943.79*** | 946.57*** |
| PseudoR$^2$ | 0.248 | 0.247 | 0.250 | 0.248 |

Note: The t-statistic is shown in parentheses.

*** is significant at 1%;

** is significant at 5%;

* is significant at 10%.

**Table 9. Heckman two-stage regression.**

| | (1) | (2) | (3) | (4) | (5) |
|---|---|---|---|---|---|
| | Rp | Rp | Rp | Rp | Rp |
| PLD | | 0.950*** | | 0.330 | |
| | | (9.714) | | (1.380) | |
| PLR | | | 1.204*** | | 0.452 |
| | | | (10.244) | | (1.430) |
| KV | | | | -1.030*** | -0.768** |
| | | | | (-2.783) | (-2.465) |
| PLD ×KV | | | | 1.172*** | |
| | | | | (2.728) | |
| PL R ×KV | | | | | 1.424** |
| | | | | | (2.512) |
| BLC | | | | | |
| PLD ×BLC | | | | | |
| PLR×BLC | | | | | |
| Size | -0.060*** | 0.704*** | 0.729*** | 0.728*** | 0.757*** |
| | (-3.808) | (7.710) | (7.935) | (7.794) | (8.059) |
| Lev | 1.189*** | -4.164*** | -4.454*** | -4.077*** | -4.410*** |
| | (13.735) | (-2.926) | (-3.128) | (-2.854) | (-3.099) |
| ROA | -0.105 | 5.103*** | 5.640*** | 5.390*** | 5.990*** |
| | (-0.326) | (5.094) | (5.716) | (5.237) | (5.907) |
| BM | -0.111*** | 0.121 | 0.144 | 0.090 | 0.118 |
| | (-5.972) | (0.814) | (0.973) | (0.596) | (0.793) |
| Top1 | -0.520*** | -1.120 | -0.853 | -1.171* | -0.885 |
| | (-5.770) | (-1.596) | (-1.222) | (-1.660) | (-1.265) |
| INST | -0.326*** | 0.438 | 0.440 | 0.439 | 0.454 |
| | (-5.214) | (0.976) | (0.983) | (0.971) | (1.010) |
| DIV | 0.014 | 0.028* | 0.026* | 0.030** | 0.028* |
| | (1.109) | (1.863) | (1.674) | (1.966) | (1.815) |
| Dual | 0.236*** | -0.267 | -0.326 | -0.241 | -0.312 |
| | (8.694) | (-0.949) | (-1.155) | (-0.852) | (-1.107) |
| FCF | -0.557*** | 0.104 | 0.145 | 0.068 | 0.125 |
| | (-4.850) | (0.136) | (0.191) | (0.089) | (0.163) |
| LAG | -0.201*** | 0.259 | 0.244 | 0.243 | 0.237 |
| | (-10.748) | (1.109) | (1.046) | (1.040) | (1.019) |
| Board | -0.617*** | 0.860 | 1.032 | 0.795 | 1.005 |
| | (1.191) | (1.422) | (1.097) | (1.167) | (0.934) |
| IMR | | -3.116* | -3.570** | -2.978 | -3.516* |
| | | (-1.710) | (-1.964) | (-1.627) | (-1.934) |
| Ind | Yes | Yes | Yes | Yes | Yes |
| Year | Yes | Yes | Yes | Yes | Yes |
| _cons | 2.311*** | -18.233*** | -18.495*** | -18.307*** | -18.739*** |
| | (6.795) | (-10.454) | (-10.478) | (-10.406) | (-10.527) |
| N | 12980 | 12944 | 12944 | 12944 | 12944 |
| WaldChi2 | 1947.30*** | 923.54*** | 937.28*** | 934.74*** | 942.95*** |

(*Continued*)

**Table 9.** (Continued)

| | (1) | (2) | (3) | (4) | (5) |
|---|---|---|---|---|---|
| | **Rp** | **Rp** | **Rp** | **Rp** | **Rp** |
| PseudoR$^2$ | 0.111 | 0.248 | 0.247 | 0.250 | 0.248 |

Note: The t-statistic is shown in parentheses.

*** is significant at 1%;

** is significant at 5%;

* is significant at 10%.

consistent with the earlier conclusions and suggest the crash of 2015 had no discernible impact.

## 5. Further analysis

### 5.1. Market reaction to the share repurchase forecast

Based on the analysis, this study found that the positive market reaction to the company's share repurchase notice is the fundamental reason for the listed company and its insiders to reduce the risk of share pledge by using the share repurchase notice. To verify this speculation, the event research method was used to explore the impact of the stock repurchase notice on the stock price under the background of the share pledge. The date of the stock repurchase announcement was taken as the event day, the event window period was [-10, + 10], and the estimation window period was [-110, -11]. The market model was used to calculate the average abnormal return (AAR) and the cumulative average abnormal return (CAAR), and the trend is shown in Fig 1. The t-test results for AAR and CAAR are shown in Table 14.

As can be seen from Fig 1, the AAR after the stock repurchase announcement is significantly greater than zero, and the CAAR gradually increases, indicating that the repurchase announcement can produce positive market effects. As seen in Table 14, the AAR on the day of the announcement is significantly positive at 1%, and the share repurchase forecast will indeed have a positive impact on the share price.

To verify the impact of the share repurchase forecast on the share price, this paper recalculated the CAAR over different windows and verified its significance. The results are shown in Table 15. The CAARs at different windows in Table 15 are significantly positive. The research results show that the company's share repurchase forecast will indeed be positive for the company's stock price, and the share repurchase forecast can alleviate the pledge risk facing the controlling shareholders to a certain extent.

### 5.2. Analysis of action mechanism

The share pledge possibly leads to the risk of increasing margin and control transfer to the superior company, and the share pledge of the controlling shareholder affects the probability

**Table 10.** Propensity score matching results.

| Psmatch2:Treatment assignment | Psmatch2: Common Support | | Total |
|---|---|---|---|
| **Psmatch2:Treatment** | **Off support** | **On support** | |
| Untreated | 15 | 7639 | 7654 |
| Treated | 2 | 5324 | 5326 |
| Total | 17 | 12963 | 4580 |

**Table 11. The balance test of propensity-to-match score.**

| Variable | N | Mean | | %bias (difference) | %reduct | t-test | |
|---|---|---|---|---|---|---|---|
| | | Treated | Control | | \|bias\| | t | p>\|t\| |
| Size | Unmatched | 22.074 | 22.365 | -23.2 | | -12.81 | 0.000 |
| | Matched | 22.074 | 22.033 | 3.3 | 85.8 | 1.80 | 0.072 |
| Lev | Unmatched | 0.405 | 0.408 | -1.6 | | -0.91 | 0.363 |
| | Matched | 0.405 | 0.399 | 2.8 | -68.7 | 1.44 | 0.150 |
| ROA | Unmatched | 0.5526 | 0.554 | -0.5 | | -0.26 | 0.797 |
| | Matched | 0.5525 | 0.561 | -2.1 | -357.1 | -1.09 | 0.277 |
| BM | Unmatched | 0.843 | 1.080 | -22.5 | | -12.44 | 0.000 |
| | Matched | 0.843 | 0.842 | 0.1 | 99.5 | 0.06 | 0.950 |
| Top1 | Unmatched | 0.338 | 0.369 | -20.9 | | -11.54 | 0.000 |
| | Matched | 0.338 | 0.336 | 1.5 | 93.0 | 0.77 | 0.440 |
| INST | Unmatched | 0.364 | 0.440 | -32.9 | | -18.36 | 0.000 |
| | Matched | 0.364 | 0.360 | 1.4 | 95.7 | 0.73 | 0.466 |
| DIV | Unmatched | 0.344 | 0.316 | 2.1 | | 1.25 | 0.213 |
| | Matched | 0.344 | 0.320 | 1.8 | 14.6 | 0.89 | 0.347 |
| Dual | Unmatched | 0.342 | 0.211 | 29.6 | | 16.83 | 0.000 |
| | Matched | 0.341 | 0.337 | 1.0 | 96.7 | 0.48 | 0.634 |
| FCF | Unmatched | 0.008 | 0.016 | -7.7 | | -4.40 | 0.000 |
| | Matched | 0.008 | 0.009 | -1.6 | 78.6 | -0.84 | 0.402 |
| LAG | Unmatched | 1.9782 | 0.206 | -29.7 | | -16.50 | 0.000 |
| | Matched | 1.9782 | 1.956 | 2.9 | 90.1 | 1.46 | 0.144 |
| Board | Unmatched | 2.0992 | 2.163 | -33.4 | | -18.66 | 0.000 |
| | Matched | 20.992 | 2.102 | -1.2 | 96.3 | -0.63 | 0.526 |

of the listed company issuing the share repurchase notice by increasing the risk of a pledge. The study follows Hu et al. [27] and Rhodes–Kropf et al. [28] and measures the pledge risk by constructing the dummy variable Pressure and continuous variable Mis. Then we analyzed the relationship between the controlling shareholder equity pledge, pledge risk, and the relationship between equity pledge and stock repurchase notice.

**5.2.1. Margin call pressure.** Referring to the research of Hu et al. [27], the intensity of the pledge risk was measured by calculating the difference between the estimated early warning price and the lowest stock price in the current period. This paper assumed that the discount rate of the pledged stock was 60%, the pledge early warning line was 160%, the pledge annual interest rate was 10%, and the interest repayment cycle was 90 days. The calculation method of interest early warning price as follow: principal and interest early warning pricePledge financing amount per share = discount rate of pledge per share; per share daily interest = annual interest rate of pledged financing amount per share / 365; principal and interest early warning price = (pledged financing amount per share + daily interest payment cycle per share) pledge early warning line ratio. The Pressure variable was constructed by comparing the estimated pledge warning line with the lowest stock price in the current period. First, the difference between the lowest share price and the expected warning stock price of each company $A_i$ was calculated. Next, the current period of $A_i$ was chosen. The average value B of the current share price gap was calculated for companies greater than 0. If the difference $A_i$ is less than 0, the company has warning pressure in the current period, and the Pressure value is 1. Meanwhile, if the company's current $A_i$ exceeds 0 but is less than B, companies also have a large warning pressure, so Pressure is also assigned a value of 1. Pressure was assigned 0 except for two cases.

**Table 12. Results of sample regression after matching.**

| | (1) | (2) | (3) | (4) |
|---|---|---|---|---|
| | Rp | Rp | Rp | Rp |
| PLD | 0.980*** | | 0.346 | |
| | (9.558) | | (1.332) | |
| PLR | | 1.277*** | | 0.477 |
| | | (10.343) | | (1.392) |
| KV | | | -1.111*** | -0.857** |
| | | | (-2.720) | (-2.501) |
| PLD ×KV | | | 1.196** | |
| | | | (2.552) | |
| PLR×KV | | | | 1.511** |
| | | | | (2.453) |
| BLC | | | | |
| PLD ×BLC | | | | |
| PLR×BLC | | | | |
| Size | 0.587*** | 0.601*** | 0.618*** | 0.636*** |
| | (9.706) | (9.919) | (9.693) | (9.921) |
| Lev | -1.954*** | -1.950*** | -1.964*** | -1.946*** |
| | (-5.505) | (-5.538) | (-5.514) | (-5.520) |
| ROA | 4.448*** | 5.049*** | 4.785*** | 5.453*** |
| | (4.201) | (4.834) | (4.384) | (5.066) |
| BM | -0.163** | -0.171** | -0.187** | -0.198** |
| | (-2.065) | (-2.141) | (-2.292) | (-2.374) |
| Top1 | -2.337*** | -2.178*** | -2.353*** | -2.196*** |
| | (-6.265) | (-5.936) | (-6.251) | (-5.948) |
| INST | -0.090 | -0.185 | -0.052 | -0.154 |
| | (-0.353) | (-0.732) | (-0.204) | (-0.604) |
| DIV | 0.047*** | 0.047*** | 0.048*** | 0.048*** |
| | (3.778) | (3.468) | (3.833) | (3.730) |
| Dual | 0.172* | 0.175* | 0.179* | 0.183* |
| | (1.659) | (1.685) | (1.715) | (1.753) |
| FCF | -1.130** | -1.253*** | -1.119** | -1.267*** |
| | (-2.433) | (-2.717) | (-2.404) | (-2.735) |
| LAG | -0.131* | -0.204*** | -0.130* | -0.204*** |
| | (-1.797) | (-2.856) | (-1.776) | (-2.852) |
| Board | -0.355 | -0.359 | -0.373 | -0.368 |
| | (-1.420) | (-1.426) | (-1.494) | (-1.463) |
| Ind | Yes | Yes | Yes | Yes |
| Year | Yes | Yes | Yes | Yes |
| cons | -16.502*** | -16.638*** | -16.663*** | -16.947*** |
| | (-10.807) | (-10.940) | (-10.778) | (-11.014) |
| N | 10740 | 10740 | 10740 | 10740 |
| WaldChi2 | 774.88*** | 790.05*** | 781.32*** | 791.23*** |
| PseudoR2 | 0.250 | 0.250 | 0.252 | 0.251 |

Note: The t-statistic is shown in parentheses.

*** is significant at 1%;

** is significant at 5%;

* is significant at 10%.

**Table 13. Regression results after removing observations in 2015.**

| | (1) | (2) | (3) | (4) |
|---|---|---|---|---|
| | Rp | Rp | Rp | Rp |
| PLD | *1.017*** | | 0.305 | |
| | *(9.902)* | | (1.232) | |
| PLR | | *1.254*** | | 0.435 |
| | | *(10.327)* | | (1.347) |
| KV | | | -1.310*** | -0.971*** |
| | | | (-3.425) | (-3.060) |
| PLD ×KV | | | *1.343*** | |
| | | | *(3.035)* | |
| PLR×KV | | | | *1.545*** |
| | | | | *(2.682)* |
| BLC | | | | |
| PLD ×BLC | | | | |
| PLR×BLC | | | | |
| Size | 0.565*** | 0.574*** | 0.608*** | 0.617*** |
| | (9.911) | (10.079) | (10.027) | (10.211) |
| Lev | -1.742*** | -1.687*** | -1.762*** | -1.685*** |
| | (-5.177) | (-5.077) | (-5.205) | (-5.054) |
| ROA | 4.670*** | 5.215*** | 5.145*** | 5.749*** |
| | (4.538) | (5.124) | (4.856) | (5.489) |
| BM | -0.116 | -0.127* | -0.146* | -0.157** |
| | (-1.606) | (-1.731) | (-1.951) | (-2.069) |
| Top1 | -2.251*** | -2.132*** | -2.272*** | -2.165*** |
| | (-6.156) | (-5.942) | (-6.139) | (-5.980) |
| INST | -0.070 | -0.176 | -0.028 | -0.136 |
| | (-0.282) | (-0.715) | (-0.110) | (-0.545) |
| DIV | 0.046*** | 0.046*** | 0.047*** | 0.047*** |
| | (3.740) | (3.425) | (3.805) | (3.690) |
| Dual | 0.180* | 0.189* | 0.188* | 0.197* |
| | (1.805) | (1.889) | (1.876) | (1.959) |
| FCF | -1.170*** | -1.306*** | -1.155** | -1.309*** |
| | (-2.609) | (-2.936) | (-2.570) | (-2.925) |
| LAG | -0.126* | -0.202*** | -0.125* | -0.203*** |
| | (-1.801) | (-2.984) | (-1.792) | (-2.998) |
| Board | -0.344 | -0.358 | -0.365 | -0.370 |
| | (-1.434) | (-1.495) | (-1.523) | (-1.545) |
| Ind | Yes | Yes | Yes | Yes |
| Year | Yes | Yes | Yes | Yes |
| _cons | -15.921*** | -15.864*** | -16.257*** | -16.342*** |
| | (-11.601) | (-11.656) | (-11.657) | (-11.797) |
| N | 11431 | 11431 | 11431 | 11431 |
| WaldChi2 | 848.04*** | 852.28*** | 859.46*** | 857.24*** |
| PseudoR2 | 0.254 | 0.251 | 0.257 | 0.253 |

Note: The t-statistic is shown in parentheses.

*** is significant at 1%;

** is significant at 5%;

* is significant at 10%.

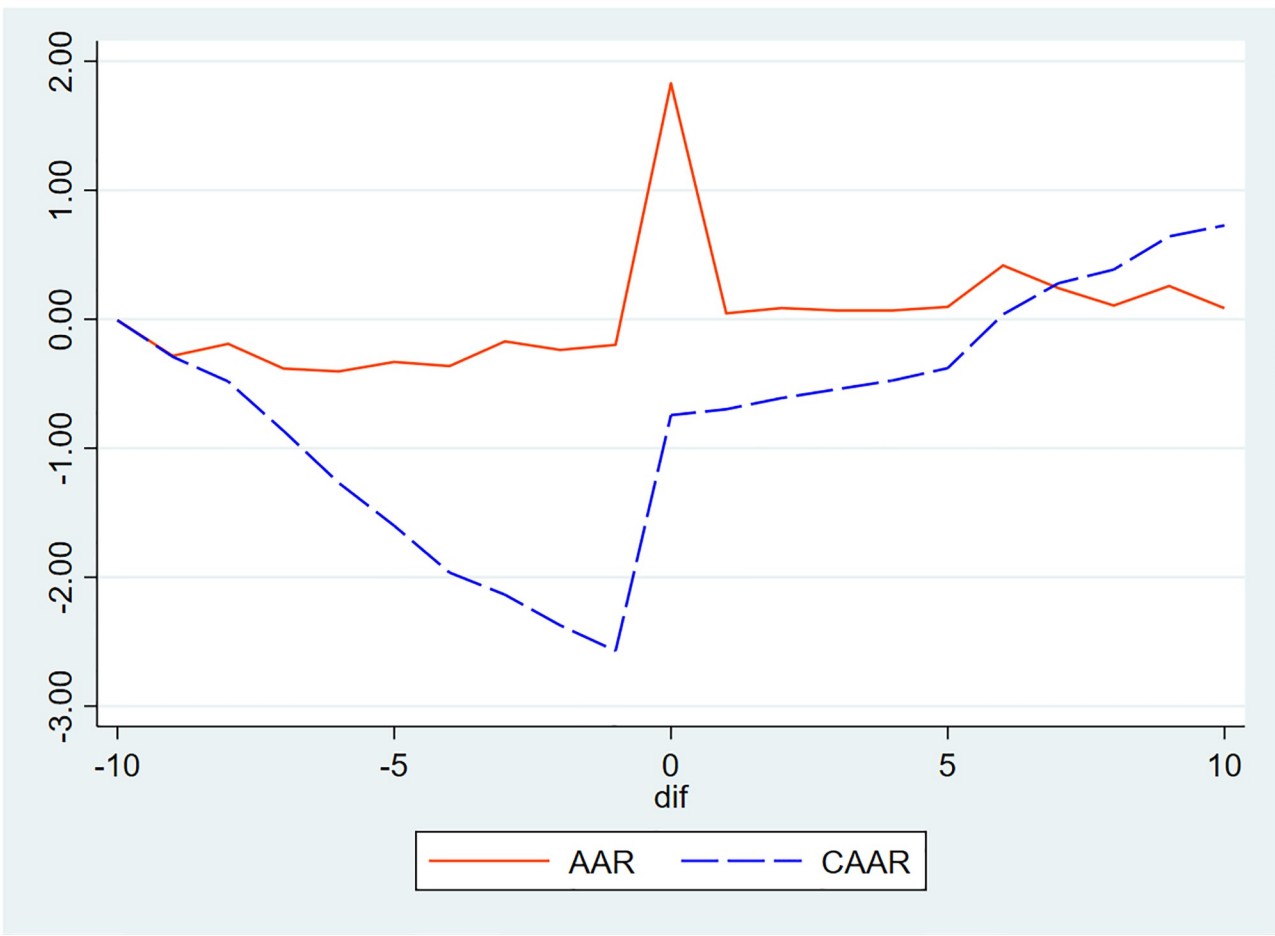

**Fig 1. Average excess return (AAR) and cumulative average excess return (CAAR) chart.**

Models (4), (5), and (6) were constructed using the gradual regression method to test the impact of early warning pressure on the controlling shareholder share pledge and stock repurchase notice. The control variables in the model (4), (5), and (6) are the same as above.

$$Rp_{i,t} = \beta_0 + \beta_1 PLD_{i,t-1}/PLR_{i,t-1} + Controls_{i,t-1} + \sum Year + \sum Ind + \varepsilon_{i,t-1} \quad (4)$$

$$Pressure_{i,t} = \beta_0 + \beta_1 PLD_{i,t-1}/PLR_{i,t-1} + Controls_{i,t-1} + \sum Year + \sum Ind + \varepsilon_{i,t-1} \quad (5)$$

$$Rp_{i,t} = \beta_0 + \beta_1 PLD_{i,t-1}/PLR_{i,t-1} + \beta_2 Pressure_{i,t} + Controls_{i,t-1} + \sum Year + \sum Ind + \varepsilon_{i,t-1} \quad (6)$$

To test the robustness of the empirical results, the Sobel test was used following the guidance of Wen et al. [29]. The regression results are shown in Table 16, and Pressure plays a partial mediating role between Rp and PLD and PLR. Current early warning pressure affects the company's current stock repurchase announcement releases. The higher the proportion of pledges, the greater the early warning pressure in the current period, and the stronger the positive relationship between share pledges and share repurchase forecast.

**Table 14. Average excess return (AAR) and cumulative average excess return (CAAR) of issuing stock repurchase plan.**

| dif | AAR | t value | CAAR | t price |
|---|---|---|---|---|
| -10 | -0.008 | -0.0955 | -0.008 | -0.0955 |
| -9 | -0.2835 | -3.3557*** | -0.2915 | -2.4142** |
| -8 | -0.1909 | -2.2036** | -0.4824 | -3.3395*** |
| -7 | -0.3823 | -4.1273*** | -0.8646 | -4.7403*** |
| -6 | -0.4045 | -3.8889*** | -1.2691 | -5.6167*** |
| -5 | -0.3312 | -3.6238*** | -1.6003 | -6.5888*** |
| -4 | -0.3633 | -3.2619*** | -1.9636 | -7.0495*** |
| -3 | -0.1717 | -1.4744 | -2.1353 | -6.9182*** |
| -2 | -0.2378 | -2.0752** | -2.3731 | -6.9146*** |
| -1 | -0.1987 | -1.4863 | -2.5717 | -6.6235*** |
| 0 | 1.8283 | 11.371*** | -0.7434 | -1.7145* |
| 1 | 0.0457 | 0.3893 | -0.6978 | -1.5367 |
| 2 | 0.0868 | 0.8862 | -0.611 | -1.3219 |
| 3 | 0.0681 | 0.7872 | -0.5429 | -1.1435 |
| 4 | 0.0679 | 0.7809 | -0.475 | -1.0016 |
| 5 | 0.0958 | 1.0305 | -0.3792 | -0.771 |
| 6 | 0.4169 | 4.5733*** | 0.0377 | 0.0747 |
| 7 | 0.2411 | 2.7085*** | 0.2788 | 0.5415 |
| 8 | 0.1061 | 1.1994 | 0.3849 | 0.7295 |
| 9 | 0.2576 | 2.9235*** | 0.6426 | 1.195 |
| 10 | 0.0851 | 0.9336 | 0.7277 | 1.3267 |

**5.2.2. Risk of stock price bias.** The degree of stock price bias reflects investors' judgment on the future development trend of the company and affects the company's future stock price. Under the background of share pledge, the higher the degree of stock price bias, the greater the probability of stock price decline, and the higher the risk of the pledge. This paper measured the pledge risk of the controlling shareholder by calculating the degree of stock price bias. Rhodes–Kropf et al. [28] studied the stock price bias (Misp) by calculating the difference between the company market value (LnM) and the estimated base value (LnV). The degree of stock price bias (Misp) was calculated as follows. First, the company data of each industry was replaced into formula (7) to estimate the regression coefficient of each industry in each $year_0$, $\beta_1$, $\beta_2$, $\beta_3$, and $\beta_4$. After that, the company data were inserted into formula (7) according to the industry of each company to obtain the estimated LnV of the base value of the company in the current year. Finally, the stock price bias (Misp) was obtained by calculating the difference between the market value (LnM) and the estimated base value (LnV). When Misp> 0, it indicates that the company's stock price is overvalued, while Misp <0 is considered undervalued. In formula (8), M is the market value of the company, the sum of the book value of non-tradable shares and the market value of tradable shares; B is the total assets of the Company; (NI)

**Table 15. CAAR test in different intervals.**

| window phase | CAAR (CAAR) | t value | window phase | CAAR (CAAR) | t value |
|---|---|---|---|---|---|
| [-5,5] | 0.8899 | 2.017** | [0,2] | 1.9607 | 7.8793*** |
| [-3,3] | 1.4207 | 3.780*** | [0,3] | 2.0288 | 7.4804*** |
| [-1,1] | 1.6753 | 6.379*** | [0,4] | 2.0967 | 7.4894*** |
| [0] | 1.8283 | 11.371*** | [0,5] | 2.1925 | 7.3837*** |

**Table 16. Mediating effect of margin call pressure.**

| | (1) | (2) | (3) | (4) | (5) | (6) |
|---|---|---|---|---|---|---|
| | Rp | Pressure | Rp | Rp | Pressure | Rp |
| PLD | 0.962*** | 3.348*** | 0.832*** | | | |
| | (9.783) | (56.561) | (6.642) | | | |
| PLR | | | | 1.213*** | 4.282*** | 0.971*** |
| | | | | (10.283) | (46.195) | (6.676) |
| Pressure | | | 0.198* | | | 0.318*** |
| | | | (1.759) | | | (2.964) |
| Size | 0.585*** | -0.031 | 0.587*** | 0.592*** | -0.033 | 0.595*** |
| | (10.543) | (-0.976) | (10.520) | (10.676) | (-1.102) | (10.632) |
| Lev | -1.845*** | 0.522*** | -1.866*** | -1.792*** | 0.697*** | -1.844*** |
| | (-5.650) | (2.820) | (-5.706) | (-5.548) | (3.986) | (-5.686) |
| ROA | 4.817*** | -1.386** | 4.908*** | 5.319*** | 0.122 | 5.352*** |
| | (4.826) | (-2.030) | (4.921) | (5.400) | (0.190) | (5.428) |
| BM | -0.109 | -0.017 | -0.108 | -0.119* | -0.028 | -0.114 |
| | (-1.562) | (-0.474) | (-1.538) | (-1.678) | (-0.795) | (-1.611) |
| Top1 | -2.197*** | 0.244 | -2.223*** | -2.086*** | 0.478*** | -2.140*** |
| | (-6.185) | (1.286) | (-6.251) | (-5.966) | (2.788) | (-6.091) |
| INST | -0.203 | 1.019*** | -0.259 | -0.295 | 0.686*** | -0.358 |
| | (-0.838) | (7.570) | (-1.067) | (-1.228) | (5.547) | (-1.490) |
| DIV | 0.046*** | 0.011 | 0.046*** | 0.046*** | 0.030 | 0.045*** |
| | (3.804) | (0.646) | (3.796) | (3.497) | (1.247) | (3.563) |
| Dual | 0.185* | 0.220*** | 0.179* | 0.192** | 0.264*** | 0.180* |
| | (1.903) | (3.765) | (1.841) | (1.972) | (4.775) | (1.840) |
| FCF | -0.995** | -0.005 | -0.998** | -1.117*** | -0.242 | -1.086** |
| | (-2.302) | (-0.021) | (-2.310) | (-2.609) | (-0.983) | (-2.529) |
| LAG | -0.127* | -0.052 | -0.133* | -0.200*** | -0.307*** | -0.194*** |
| | (-1.872) | (-1.276) | (-1.957) | (-3.005) | (-8.344) | (-2.892) |
| Board | -0.338 | -0.077 | -0.341 | -0.345 | -0.094 | -0.345 |
| | (-1.454) | (-0.580) | (-1.462) | (-1.481) | (-0.732) | (-1.478) |
| Ind | Yes | Yes | Yes | Yes | Yes | Yes |
| Year | Yes | Yes | Yes | Yes | Yes | Yes |
| _cons | -16.360*** | -1.562** | -16.398*** | -16.313*** | -0.772 | -16.414*** |
| | (-12.104) | (-2.225) | (-12.081) | (-12.144) | (-1.179) | (-12.137) |
| N | 12944 | 12980 | 12944 | 12944 | 12980 | 12944 |
| WaldChi2 | 923.82*** | 4187.17*** | 921.13*** | 937.28*** | 2471.95*** | 933.04*** |
| PseudoR2 | 0.248 | 0.357 | 0.248 | 0.246 | 0.288 | 0.247 |
| Sobel test | | 0.738 | | | 2.387** | |

Note: The t-statistic is shown in parentheses.

*** is significant at 1%;

** is significant at 5%;

* is significant at 10%.

+ is the absolute value of the Company net income; I ($<0$) is a binary variable, when NI $<0$, meaning the Company net income is negative, I ($<0$) is 1, otherwise I ($<0$) is 0; Lev is the ratio of total liabilities to total assets; and V is the underlying value of the company.

To test whether the degree of stock price bias plays an intermediary role between the controlling shareholders' share pledge and the stock repurchase notice, this paper constructs the models (8), (9) and (10). Model (8), (9) and (10) control variables are consistent with the earlier models.

$$LnM_{i,t} = \beta_0 + \beta_1 LnB_{i,t} + \beta_2 Ln(\text{NI})_{i,t}^{+} + \beta_3 I_{(<0)} Ln(\text{NI})_{i,t}^{+} + \beta_4 Lev_{i,t} + \varepsilon_{i,t} \tag{7}$$

$$Rp_{i,t} = \beta_0 + \beta_1 PLD_{i,t-1}/PLR_{i,t-1} + \beta_2 Misp_{i,t-1} Controls_{i,t-1} + \sum Year + \sum Ind + \varepsilon_{i,t-1} \tag{8}$$

$$Rp_{i,t} = \beta_0 + \beta_1 PLD_{i,t-1}/PLR_{i,t-1} + Controls_{i,t-1} + \sum Year + \sum Ind + \varepsilon_{i,t-1} \tag{9}$$

$$Misp_{i,t-1} = \beta_0 + \beta_1 PLD_{i,t-1}/PLR_{i,t-1} + Controls_{i,t-1} + \sum Year + \sum Ind + \varepsilon_{i,t-1} \tag{10}$$

The regression results of the models (9), (10), and (11) are shown in Table 17. From Table 17, we see the regression coefficient between Rp and PLD and PLR is significantly positive at 1%, that between Misp and PLD and PLR is significantly negative at 1%, Rp and PLD and PLR, and the coefficient between Rp and Misp is significantly negative. Therefore, it can be inferred that the share pledge of the controlling shareholder will increase the downward pressure on the stock price, and the worse the underestimate of the stock price, the higher the probability of the listed company using the stock repurchase notice to pull up the stock price to reduce the pledge risk. The risk of stock price bias plays an intermediary role in the controlling shareholder's pledge and share repurchase notice.

## 5.3 Economic consequences of real repurchase under the background of share pledge

Stock repurchase notice can not only be used as a means of market value management to convey the real information of the company to the capital market, but also as a tool for the controlling shareholders to manipulate stock prices to mislead investors. The former is based on the real value of the company and is conducive to the long-term development of the enterprise, while the latter belongs to the selfish behavior of the controlling shareholder, which damages the interests of minority shareholders. Therefore, analyzing the economic consequences of share repurchase under the background of pledge is helpful to deepen the understanding of the relationship between the controlling shareholder's equity pledge and stock repurchase notices.

The return on total assets (ROA) was used to measure business performance, and the Tobin Q (TobinQ) was used to measure business value. According to the current repurchase structure, the analysis used control variables including ARp to measure the current repurchase of listed companies, enterprise-scale (Size), leverage ratio (Lev), cash flow asset utilization (Cashflow), the largest shareholder shareholding ratio (Top1), institutional investors (INST), enterprise free cash flow (FCF), the top three executives' total compensation (lntop3a), and the two rights separation degree (Sep). The flow of assets utilization (Cashflow) is equal to the net cash flow generated by operating activities divided by the total assets, the three total compensation (lntop3a) is equal to the total compensation of the top three executives, the degree of separation (Sep) is the difference between the control and ownership of the controlling shareholder,

**Table 17. Mediating effect of stock misprice.**

| | (1) | (2) | (3) | (4) | (5) | (6) |
|---|---|---|---|---|---|---|
| | Rp | Misp | Rp | Rp | Misp | Rp |
| PLD | *0.953*** | *0.016*** | *0.963*** | | | |
| | *(9.650)* | *(2.898)* | *(9.747)* | | | |
| PLR | | | | *1.213*** | *0.020** | *1.218*** |
| | | | | *(10.171)* | *(2.394)* | *(10.208)* |
| Misp | | | *-0.550*** | | | *-0.515*** |
| | | | *(-3.600)* | | | *(-3.420)* |
| Size | 0.589*** | 0.042*** | 0.619*** | 0.596*** | 0.042*** | 0.623*** |
| | (10.515) | (11.212) | (10.620) | (10.636) | (11.220) | (10.722) |
| Lev | -1.835*** | 0.017 | -1.787*** | -1.782*** | 0.018 | -1.729*** |
| | (-5.570) | (0.828) | (-5.367) | (-5.469) | (0.891) | (-5.251) |
| ROA | 4.781*** | 0.905*** | 5.528*** | 5.283*** | 0.910*** | 5.956*** |
| | (4.781) | (9.959) | (5.412) | (5.350) | (10.017) | (5.892) |
| BM | -0.105 | -0.101*** | -0.180** | -0.116 | -0.101*** | -0.187** |
| | (-1.514) | (-20.874) | (-2.199) | (-1.637) | (-20.881) | (-2.259) |
| Top1 | -2.191*** | -0.257*** | -2.345*** | -2.078*** | -0.256*** | -2.229*** |
| | (-6.108) | (-13.512) | (-6.441) | (-5.887) | (-13.436) | (-6.219) |
| INST | -0.224 | 0.512*** | 0.047 | -0.316 | 0.511*** | -0.058 |
| | (-0.919) | (36.206) | (0.182) | (-1.307) | (36.139) | (-0.227) |
| DIV | 0.045*** | 0.010*** | 0.050*** | 0.045*** | 0.010*** | 0.050*** |
| | (3.706) | (6.333) | (4.228) | (3.429) | (6.406) | (3.901) |
| Dual | 0.180* | 0.031*** | 0.200** | 0.185* | 0.031*** | 0.203** |
| | (1.837) | (5.000) | (2.037) | (1.885) | (5.050) | (2.064) |
| FCF | -0.967** | 0.047* | -0.940** | -1.096** | 0.046* | -1.074** |
| | (-2.210) | (1.797) | (-2.155) | (-2.529) | (1.754) | (-2.488) |
| LAG | -0.130* | 0.067*** | -0.100 | -0.201*** | 0.066*** | -0.173** |
| | (-1.890) | (15.215) | (-1.444) | (-2.990) | (15.085) | (-2.553) |
| Board | -0.314 | -0.073*** | -0.328 | -0.322 | -0.073*** | -0.336 |
| | (-1.343) | (-5.003) | (-1.403) | (-1.372) | (-5.027) | (-1.434) |
| Ind | Yes | Yes | Yes | Yes | Yes | Yes |
| Year | Yes | Yes | Yes | Yes | Yes | Yes |
| _cons | -16.441*** | -0.902*** | -17.134*** | -16.419*** | -0.899*** | -17.033*** |
| | (-12.048) | (-11.030) | (-12.251) | (-12.102) | (-11.027) | (-12.299) |
| N | 12644 | 12679 | 12644 | 12644 | 12679 | 12644 |
| WaldChi2 | 901.62*** | 121.83*** | 904.48*** | 915.02*** | 121.26*** | 918.46*** |
| PseudoR2/R2 | 0.245 | 0.278 | 0.247 | 0.243 | 0.278 | 0.245 |
| Sobel test | | -1.900* | | | -1.790* | |

Note: The t-statistic is shown in parentheses.

*** is significant at 1%;

** is significant at 5%;

* is significant at 10%.

the calculation method and definition of the remaining control variables are the same as in Table 3.

Model (11) was constructed to test the impact of real repurchase on business performance and enterprise value under the background of share pledges. The results of the regression are

**Table 18. Economic consequences of real stock repurchases.**

| | (1) | (2) | (3) | (4) |
|---|---|---|---|---|
| | ROA | ROA | TobinQ | TobinQ |
| ARP | 0.007*** | 0.007*** | -0.119 | -0.087 |
| | (3.153) | (2.636) | (-1.556) | (-0.877) |
| PLR | 0.002** | | 0.334*** | |
| | (2.208) | | (5.519) | |
| ARP×PLR | *-0.011*** | | *-0.241** | |
| | *(-2.834)* | | *(-1.757)* | |
| PLD | | 0.003*** | | 0.272*** |
| | | (4.700) | | (7.024) |
| ARP×PLD | | *-0.007*** | | *-0.199** |
| | | *(-2.320)* | | *(-1.810)* |
| Lev | -0.080*** | -0.081*** | -1.679*** | -1.681*** |
| | (-36.426) | (-36.707) | (-12.486) | (-12.567) |
| Size | 0.001* | 0.001** | -0.666*** | -0.663*** |
| | (1.941) | (2.091) | (-23.226) | (-22.980) |
| Cashflow | 0.227*** | 0.227*** | 2.530*** | 2.508*** |
| | (33.548) | (33.617) | (7.338) | (7.299) |
| Top1 | 0.013*** | 0.013*** | -0.127 | -0.157 |
| | (5.265) | (5.376) | (-0.972) | (-1.207) |
| INST | 0.011*** | 0.012*** | 1.094*** | 1.129*** |
| | (6.472) | (6.758) | (10.811) | (11.226) |
| lntop3a | 0.012*** | 0.012*** | 0.324*** | 0.322*** |
| | (20.402) | (20.497) | (11.405) | (11.336) |
| FCF | 0.024*** | 0.024*** | 0.521** | 0.533*** |
| | (6.400) | (6.414) | (2.535) | (2.589) |
| Sep | -0.000** | -0.000*** | -0.014*** | -0.014*** |
| | (-2.471) | (-2.612) | (-7.576) | (-7.425) |
| Ind | Yes | Yes | Yes | Yes |
| Year | Yes | Yes | Yes | Yes |
| _cons | -0.120*** | -0.122*** | 12.409*** | 12.347*** |
| | (-13.012) | (-13.277) | (18.772) | (18.565) |
| N | 12520 | 12520 | 12520 | 12520 |
| R2 | 0.352 | 0.353 | 0.347 | 0.348 |

Note: The t-statistic is shown in parentheses.

*** is significant at 1%;

** is significant at 5%;

* is significant at 10%.

shown in Table 18. Under the background of share pledge, the enterprise share repurchase will reduce the business performance and damage the enterprise value, so it is inferred that the stock repurchase notice may be a means for the controlling shareholder to manipulate the stock price and alleviate the pledge risk.

$$ROA_{i,t}/TobinQ_{i,t} = \beta_0 + \beta_1 ARp_{i,t} + Controls_{i,t} + \sum Year + \Sigma Ind + \varepsilon_{i,t} \qquad (11)$$

## 6. Conclusions

This study used Chinese A-share listed companies from 2012 to 2019 as the research sample. Relevant data on stock repurchase forecasts were collected, and the study then used a comprehensive variety of methods to explore whether the share pledge of the controlling shareholder impacts the release of stock repurchase notices. This paper analyzed the influence of information disclosure quality on the controlling shareholder share pledge and stock repurchase notice and further explored the mechanism between share pledge and stock repurchase notice. To test for the real motivation of listed companies to release the share repurchase notice under the background of share pledge, we compared the impact of the real repurchase on the business performance and company value under different situations, and we can make the following four key conclusions.

First, pledge risk management may be the potential motivation for the listed company to issue a share repurchase notice. When the controlling shareholder of the listed company has an share pledge, the probability of the listed company issuing a share repurchase notice is significantly increased.

Second, the quality of information disclosure will affect the relationship between share pledges and share repurchase notices. The worse the quality of information disclosure, the higher the probability of the listed company and its insiders using share repurchase notice to raise the share price, and the closer the relationship between share pledge and share repurchase notice.

Third, the stock repurchase forecast positively affects the stock price in the short term, and the pledge risk brought by the share pledge is an important motivation for the listed companies to issue the share repurchase notice.

Fourth, under the background of the share pledge, the enterprises' repurchase behavior will harm the business performance and company value. The higher the share pledge ratio, the more obvious the negative effect of real repurchase.

From these research findings, we can state that the share pledge of the controlling shareholder impacts the stock repurchase notice of listed companies. Because the share repurchase notice can quickly increase the stock price, when the company faces the pledge risk, the listed company and its insiders will use the share repurchase notice for the pledge risk management. The study found that a company with the share pledge of the controlling shareholder's real repurchase behavior will negatively impact the company's development.

The conclusion of this paper has certain reference value for the regulatory authorities and outside investors. For the Regulators, how to supervise the stock repurchase of listed companies and avoid the manipulation of the stock price by enterprises, which will harm the interests of investors, will be an important task in the next stage. While actively encouraging the stock repurchase of listed companies, the regulatory authorities should also constantly improve the relevant regulatory policies. For investors, to be rational about the company's share repurchase notice, blind to the company announced as good news may fall into the trap of large shareholders, suffer huge losses.

The paper has the following limitations: firstly, the paper mainly studies the stock repurchase notice, and little research on stock repurchase. In the future, research can be conducted from the perspective of stock repurchase; Secondly, The current research sample is a-share listed companies, the data on the Second Board and STAR Market may draw different conclusions.

## Author Contributions

**Data curation:** Limei Cao.

**Investigation:** Xiaoyun Zhu.

**Methodology:** Xiaoyun Zhu.

**Project administration:** Jiani Xie.

**Resources:** Jiani Xie.

**Writing – original draft:** Limei Cao, Xiaoyun Zhu.

**Writing – review & editing:** Jiani Xie.

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
