## [Decision Letter · Decision Letter 0]

9 Jun 2024

PONE-D-24-15737Controlling Shareholders’ Share Pledge and Share Repurchase Notices of Listed CompaniesPLOS ONE

Dear Dr. Cao,

Thank you for submitting your manuscript to PLOS ONE. After careful consideration, we feel that it has merit but does not fully meet PLOS ONE’s publication criteria as it currently stands. Therefore, we invite you to submit a revised version of the manuscript that addresses the points raised during the review process.

We look forward to receiving your revised manuscript.

Kind regards,

Amira M. Idrees, Professor

Academic Editor

PLOS ONE

Journal Requirements:

4. Thank you for stating the following financial disclosure: "Guangdong Provincial Department of Education Innovation Team Project "Big Data Audit Research Team" (2022WCXTD009); General Project of Philosophy and Social Science Planning in Guangdong Province（NO.GD23CGL27); General projects of the National Social Science Fund (23BGL053)."

Reviewers' comments:

Reviewer's Responses to Questions

**Comments to the Author**

1. Is the manuscript technically sound, and do the data support the conclusions?

Reviewer #1: Partly

2. Has the statistical analysis been performed appropriately and rigorously? 

Reviewer #1: Yes

3. Have the authors made all data underlying the findings in their manuscript fully available?

Reviewer #1: Yes

4. Is the manuscript presented in an intelligible fashion and written in standard English?

Reviewer #1: No

5. Review Comments to the Author

Reviewer #1: Comment 1: The introduction lacks a comprehensive review of existing literature on stock repurchases, especially in the context of China's market. More background information is needed to situate the study within the existing body of knowledge.

Comment 2: The specific research objectives and hypotheses should be clearly articulated. It's not immediately apparent what the authors are trying to investigate beyond the relationship between share pledges and stock repurchase notices.

Comment 3: The narrative lacks clarity and coherence, making it challenging for the reader to follow the argument seamlessly. The prose is convoluted, with a plethora of ideas and assertions that are not adequately linked. The author transitions abruptly between topics without clear signposting or logical flow, which hampers comprehension and detracts from the persuasiveness of the analysis.

Comment 4: The hypothesis suffers from inadequate referencing and citation practices. The text frequently contains placeholder texts such as "Error! Reference source not found" where citations are intended. This oversight significantly diminishes the scholarly credibility of the work and suggests a lack of attention to detail in its preparation.

Comment 5: The choice of Chinese A-share listed companies as the research sample is reasonable given the focus on a specific market context. However, the section lacks detailed justification for the specific timeframe (2012-2019) chosen for the study. The rationale behind excluding pre-2012 data, aside from the change in share approval requirements, could be more explicitly addressed to ensure transparency and completeness of the research process.

Comment 6: The conclusions would benefit from a discussion of methodological limitations and potential alternative explanations for the observed relationships. Addressing these considerations would strengthen the robustness and generalizability of the conclusions.

Comment 7: The conclusions would be more comprehensive with a discussion of study limitations and suggestions for future research. Highlighting limitations, such as data constraints or potential biases, would provide context for interpreting the findings and guide future research efforts in this area.

6. PLOS authors have the option to publish the peer review history of their article (what does this mean?). If published, this will include your full peer review and any attached files.

Reviewer #1: No

---

## [Author Response · Author response to Decision Letter 0]

7 Jul 2024

Review response

Reviewer #1:

Comment 1: The introduction lacks a comprehensive review of existing literature on stock repurchases, especially in the context of China's market. More background information is needed to situate the study within the existing body of knowledge.

Response

We thank the Editor for this suggestion，A literature review has been added in the introduction. In response to the Editor’s comments, we have revised the introduction on page 1. For your convenience, we excerpt the relative revision as follows:

The use of share buybacks by companies can send a signal to the outside world that share prices are undervalued, improve the capital structure of the company, and increase enterprise value [1]. The more confident the manager is to the enterprise, the higher the probability of stock repurchase of the listed company. But the prevalence of agency problems makes it possible that share repurchases may be alienated as a way for insiders to manipulate earnings [2], controlling shareholders and executives may also issue stock repurchase alerts for personal gain and encroachment on the company, and companies are more likely to buy back shares when executive compensation is linked to earnings per share [3]. Li and He took the case of share repurchase in the open market of our country's stock market after 2005 as the research object, and find that the information content in the repurchase announcement conforms to the undervaluation hypothesis, there are opportunities to achieve large shareholders arbitrage phenomenon [4]. Share repurchase announcements are accompanied by a reduction in long-term stock returns and insider selling [5].

Comment 2: The specific research objectives and hypotheses should be clearly articulated. It's not immediately apparent what the authors are trying to investigate beyond the relationship between share pledges and stock repurchase notices.

Response

We thank the Editor for this suggestion，The specific research objectives and hypotheses has been added in the introduction. In response to the Editor’s comments, we have revised the introduction on page 2. For your convenience, we excerpt the relative revision as follows:

The main issue of this paper is: Will the pledge of majority shareholders increase the probability that the listed companies will issue share repurchase plans to convey the signal of underpricing to the market, and how will the market react to the repurchase plans? After the release of the plan, whether the pledge of major shareholders will affect the company's ultimate repurchase implementation? Will the execution of share repurchases affect the long-term return of the company's stock? This paper intends to explore these issues in depth.

Comment 3: The narrative lacks clarity and coherence, making it challenging for the reader to follow the argument seamlessly. The prose is convoluted, with a plethora of ideas and assertions that are not adequately linked. The author transitions abruptly between topics without clear signposting or logical flow, which hampers comprehension and detracts from the persuasiveness of the analysis.

Response

We thank the Editor for this suggestion，Hypothesis 1 is the main hypothesis of the paper, hypothesis 2 and hypothesis 3 test its regulatory effects, the two regulatory effects are not linked, put together really complex, make readers difficult to understand, so we adjusted our hypothesis , remove hypothesis 3. we have revised the Theoretical analysis and research hypothesis on page 4. 

Comment 4: The hypothesis suffers from inadequate referencing and citation practices. The text frequently contains placeholder texts such as "Error! Reference source not found" where citations are intended. This oversight significantly diminishes the scholarly credibility of the work and suggests a lack of attention to detail in its preparation.

Response

We thank the Editor for this suggestion, the format of the referenced references has been checked and modified Within the entire paper.

Comment 5: The choice of Chinese A-share listed companies as the research sample is reasonable given the focus on a specific market context. However, the section lacks detailed justification for the specific timeframe (2012-2019) chosen for the study. The rationale behind excluding pre-2012 data, aside from the change in share approval requirements, could be more explicitly addressed to ensure transparency and completeness of the research process.

Response

We thank the Editor for this suggestion. The rationale behind 2020 data has been added in the Samples and Data Sources. In response to the Editor’s comments, we have revised on page 4. For your convenience, we excerpt the relative revision as follows:

China's economy, stock market and corporate repurchase activity were frequently shut down in 2020 due to the covid-19 pandemic, so the sample period did not include 2020 and beyond.

Comment 6: The conclusions would benefit from a discussion of methodological limitations and potential alternative explanations for the observed relationships. Addressing these considerations would strengthen the robustness and generalizability of the conclusions.

Response

We thank the Editor for this suggestion. According to the thesis research proposition and the goal of the research part of the conclusions were revised, increased the significance of application. In response to the Editor’s comments, we have revised on page 25. For your convenience, we excerpt the relative revision as follows:

The conclusion of this paper has certain reference value for the regulatory authorities and outside investors. For the Regulators, how to supervise the stock repurchase of listed companies and avoid the manipulation of the stock price by enterprises, which will harm the interests of investors, will be an important task in the next stage. While actively encouraging the stock repurchase of listed companies, the regulatory authorities should also constantly improve the relevant regulatory policies. For investors, to be rational about the company's share repurchase notice, blind to the company announced as good news may fall into the trap of large shareholders, suffer huge losses.

Comment 7: The conclusions would be more comprehensive with a discussion of study limitations and suggestions for future research. Highlighting limitations, such as data constraints or potential biases, would provide context for interpreting the findings and guide future research efforts in this area.

Response

We thank the Editor for this suggestion. The limitation of the paper and the field and direction of future research have been added, In response to the Editor’s comments, we have revised on page 26. For your convenience, we excerpt the relative revision as follows:

The paper has the following limitations: firstly, the paper mainly studies the stock repurchase notice, and little research on stock repurchase. In the future, research can be conducted from the perspective of stock repurchase; Secondly, the current research sample is a-share listed companies, the data on the Second Board and STAR Market may draw different conclusions.

---

## [Decision Letter · Decision Letter 1]

29 Jul 2024

Controlling Shareholders’ Share Pledge and Share Repurchase Notices of Listed Companies

PONE-D-24-15737R1

Dear Authors,

We’re pleased to inform you that your manuscript has been judged scientifically suitable for publication and will be formally accepted for publication once it meets all outstanding technical requirements.

Kind regards,

Amira M. Idrees, Professor

Academic Editor

PLOS ONE

Additional Editor Comments (optional):

Reviewers' comments:

Reviewer's Responses to Questions

**Comments to the Author**

1. If the authors have adequately addressed your comments raised in a previous round of review and you feel that this manuscript is now acceptable for publication, you may indicate that here to bypass the “Comments to the Author” section, enter your conflict of interest statement in the “Confidential to Editor” section, and submit your "Accept" recommendation.

Reviewer #1: All comments have been addressed

2. Is the manuscript technically sound, and do the data support the conclusions?

Reviewer #1: Yes

3. Has the statistical analysis been performed appropriately and rigorously? 

Reviewer #1: Yes

4. Have the authors made all data underlying the findings in their manuscript fully available?

Reviewer #1: Yes

5. Is the manuscript presented in an intelligible fashion and written in standard English?

Reviewer #1: Yes

6. Review Comments to the Author

Reviewer #1: I have carefully reviewed the revised manuscript and noted that the authors have effectively addressed all the necessary corrections and concerns raised during the initial review. The modifications have significantly improved the clarity and quality of the paper. Given the thoroughness of their revisions and the enhanced strength of the research presented, I recommend that the paper be accepted for publication. The authors have demonstrated a strong commitment to scientific rigor and have provided valuable contributions to the field. Therefore, I believe this paper meets the high standards required for publication in our esteemed journal.

7. PLOS authors have the option to publish the peer review history of their article (what does this mean?). If published, this will include your full peer review and any attached files.

Reviewer #1: No

---

## [Editor Report · Acceptance letter]

9 Aug 2024

PONE-D-24-15737R1 

PLOS ONE

Dear Dr. Cao, 

I'm pleased to inform you that your manuscript has been deemed suitable for publication in PLOS ONE. Congratulations! Your manuscript is now being handed over to our production team.

Kind regards, 

on behalf of

Prof. Amira M. Idrees 

Academic Editor

PLOS ONE